cognition/behaviour/human-computer interaction

virtual reality, navigation, decision science, networked virtual environment, crowd-source, cartography

**Author for correspondence:**
Hantao Zhao
e-mail: hantao.zhao@gess.ethz.ch

†These authors contributed equally to this study.

# The interaction between map complexity and crowd movement on navigation decisions in virtual reality

Hantao Zhao[1,†], Tyler Thrash[1,3,4,†], Armin Grossrieder[1,5], Mubbasir Kapadia[6], Mehdi Moussaïd[7], Christoph Hölscher[1] and Victor R. Schinazi[1,2]

[1]Chair of Cognitive Science, and [2]Institute of Cartography and Geoinformation, ETH Zürich, Zürich, Switzerland
[3]Geographic Information Visualization and Analysis, and [4]Digital Society Initiative, University of Zürich, Zürich, Switzerland
[5]School of Engineering, ZHAW, Winterthur, Switzerland
[6]Department of Computer Science, Rutgers University, New Brunswick, NJ, USA
[7]Center for Adaptive Rationality, Max Planck Institute for Human Development, Berlin, Germany

 HZ, 0000-0003-0398-3842; TT, 0000-0002-3011-7029;
MK, 0000-0002-3501-0028; MM, 0000-0002-1268-6332;
CH, 0000-0002-7520-9146; VRS, 0000-0002-2345-2806

A carefully designed map can reduce pedestrians' cognitive load during wayfinding and may be an especially useful navigation aid in crowded public environments. In the present paper, we report three studies that investigated the effects of map complexity and crowd movement on wayfinding time, accuracy and hesitation using both online and laboratory-based networked virtual reality (VR) platforms. In the online study, we found that simple map designs led to shorter decision times and higher accuracy compared to complex map designs. In the networked VR set-up, we found that co-present participants made very few errors. In the final VR study, we replayed the traces of participants' avatars from the second study so that they indicated a different direction than the maps. In this scenario, we found an interaction between map design and crowd movement in terms of decision time and the distributions of locations at which participants hesitated. Together, these findings can help the designers of maps for public spaces account for the movements of real crowds.

## 1. Introduction

Wayfinding refers to the decision-making component of navigation behaviour [1, pp. 257–294]. There are several sources of spatial

information in an observer's environment that may facilitate decision-making, and people can flexibly switch from one source to another [2]. People may even convey spatial information to each other, either as part of the observer's immediate environment or via signage and maps (i.e. geographical information services). For the present study, we investigated the effects of cues in the immediate environment and from other people on decision-making during navigation through a virtual airport terminal.

Often in public spaces, the behaviour of other pedestrians can influence an observer's spatial decision-making by providing environmental cues [3–5]. For example, computer-controlled agents may cause a human navigator to hesitate while searching for an exit from a virtual tunnel [6]. Humans also have the tendency to follow other people within a small group or crowd [6–8], especially in stressful conditions. For example, Moussaïd and colleagues [9] found that individuals were more likely to follow others as a crowd during a stressful (virtual) evacuation than during a relatively calm wayfinding task. However, other studies have suggested that this type of following behaviour does not always occur during wayfinding [10], even during an emergency evacuation [11]. Such following behaviour has been previously defined as 'herding', although Haghani and colleagues [12] suggest that this term is not consistently used in the literature. Here, we define 'following behaviour' as conforming to the actions of the crowd.

Following behaviour may be similar to other socially contagious behaviours such as joint visual attention [7] and judgement propagation [13]. This type of conformity may also be considered a result of group or collective intelligence. On the one hand, this intelligence can generally lead to more accurate [14] and creative decisions [15]. On the other hand, previous research has found that collective judgement can supersede individuals' abilities to make accurate decisions [16], possibly representing mindless acquiescence and lack of original thought [8].

Typically, sources of spatial information in the immediate environment can be contrasted with maps and signage because of their purposeful design and reliable data sources [17]. Signage design can improve navigation efficiency and the accuracy of wayfinding decisions in public spaces and indoor environments [18–20]. In particular, maps can be designed to visually convey relevant and accurate geographical information such as landmarks [21] and orientations [22]. Indeed, visual variables [23] such as colour and contrast [24] can be used to represent geographical information in a perceptually salient manner [25] and to facilitate the user's understanding of this information [26]. In general, well-designed maps may improve navigation efficiency [27] and reduce cognitive load [28,29], but inaccurate or misleading maps may result in getting lost [30,31] or casualties [32]. In addition, the presence of landmarks on a map can improve user satisfaction [33] and help users learn the layout of the environment [34]. However, some researchers have shown that map design did not significantly affect wayfinding performance in real [35] or virtual environments [36].

Virtual reality (VR) allows for experimental designs that would be difficult if not impossible to recreate in the real world because of practical (e.g. cost) or ethical (e.g. safety) issues [37,38]. Compared to most laboratory studies, VR also provides relatively high ecological validity [37,39]. While navigation behaviour in VR can be unrealistic in terms of speed and collision avoidance [40], an appropriate control interface and training can mediate these difficulties [41–43].

Multi-user frameworks [9,10,44–46] and online studies [47–49] can extend traditional laboratory experiments by enabling the study of large groups and expediting data collection. For example, Moussaïd and colleagues [9] implemented a multi-user VR framework using a study of collective navigation and following behaviour during a stressful evacuation. In addition, the costs of a study can be reduced using online crowd-sourcing platforms such as Amazon Mechanical Turk (AMT) [47]. This web service allows researchers to give tasks to participants that require human intelligence [50]. For example, scientists in artificial intelligence have used AMT as a data collection scheme to construct image datasets [49].

In the present paper, we describe three experiments that employ a combination of multi-user and online platforms in order to investigate the impact of crowd movement and map design on wayfinding decisions. Across these three studies, we shift the focus of spatial cognition from individual decision-making in isolation to collective decision-making in social environments. Altogether, these findings may be used for the development of new guidelines for map design and public information services.

- (i) **Study 1:** We first studied the effect of map complexity in isolation. Towards this end, using AMT, we investigated whether a complex map would delay people's wayfinding decisions and reduce accuracy in a virtual airport.
- (ii) **Study 2:** In the second study, we then tested for the effect of map complexity in a social environment. Here, we conducted a multi-user study in a networked desktop VR set-up to collect the movement trajectories of a large group of participants in a simple virtual environment. Specifically, participants were asked to turn left or right at a Y-shaped

intersection using a map that varied in complexity across trials. The trajectories and map design from the second study were also critical for the systematic variation of crowd movement in the subsequent study.

(iii) **Study 3:** In the third study, participants navigated through the same virtual environment among a crowd that was based on the trajectories from the second study, except that the crowd and the map indicated different directions. This conflict between the directions indicated by the map and the crowd allowed us to disentangle these effects on decision-making during wayfinding.

# 2. Study 1

The purpose of the first study was to examine the effect of map design on the time required to make a navigation decision at an intersection. The study was conducted using AMT because many participants could be recruited within a short period of time.

## 2.1. Participants

Participants from AMT were selected based on two conditions. First, participants had to have finished at least three other AMT tasks so that they have experience in completing AMT tasks. Second, the participants could not be located in the USA. The latter criterion is for excluding US citizens who have been to O'Hare Airport. In total, we recruited 182 participants. Thirty-seven participants were excluded from analysis because they either did not complete the experiment or zoomed in or out during the video. Participants were not allowed to perform any zooming operation because changing the zoom level would influence their performance on the task. Of the remaining 145 participants (mean age = 36.31, range = 22–73), there were 93 men and 52 women. Participants were compensated with a base reward of 1.5 USD and two possible bonuses. Participants were given a bonus of 0.8 points if they answered correctly, and a bonus of between 0.0 and 0.2 points that depended on the time between the start of the trial and their final decision. The sum of these two bonuses was multiplied by the base reward, averaged over trials and added to the base reward. Overall, participants were compensated between 1.5 and 3 USD (mean = 2.78) for approximately 20 min of participation.

## 2.2. Materials

We generated an abstract map based on the real map of O'Hare Airport Terminal 2 (figure 1a). Airport terminals were chosen by previous researchers because of their size and complexity [51–53]. This specific terminal was chosen because of its Y-shape and other design elements (e.g. annotations, gate numbers). Indeed, the Y-shaped environment allows us to conduct studies in which only a binary choice can be made. Four types of maps were created from the real map, including a simple map design with limited annotations and symbols, a complex map with rich annotations and symbols, a mirrored version (left to right) of the simple map, and a mirrored version of the complex map (figure 1). We converted the three-dimensional map into two-dimensional map because the original maps from O'Hare represent multiple floors that we did not have in the virtual environment. Without these additional floors, the maps look three-dimensional but do not provide any additional information. The virtual environment was developed using the Unity 3D game engine [54] (figure 2). Videos of movement through the corridor were recorded from the perspective of an avatar that was controlled by a researcher. The virtual avatar was animated using the ADAPT [55] framework, which facilitates the designing and authoring of virtual human characters in the Unity game engine. In each video, the avatar walked from a first person perspective at full speed (1.3 m s$^{-1}$) from the bottom part of the main corridor to the forking of the Y-shaped intersection (figure 2). The avatar stopped moving in front of the wall containing the map. We created four videos that were identical except for the map design on the wall of the intersection. Each video lasted 30 s.

## 2.3. Procedure

Participants were first asked to digitally sign a consent form, to read an information page that introduced the task, and to complete a training trial that familiarized them with the main task. The main task required participants to watch each video and decide whether to turn left or right at the Y-shaped intersection as quickly as possible. Participants were instructed that faster decision-making resulted in higher

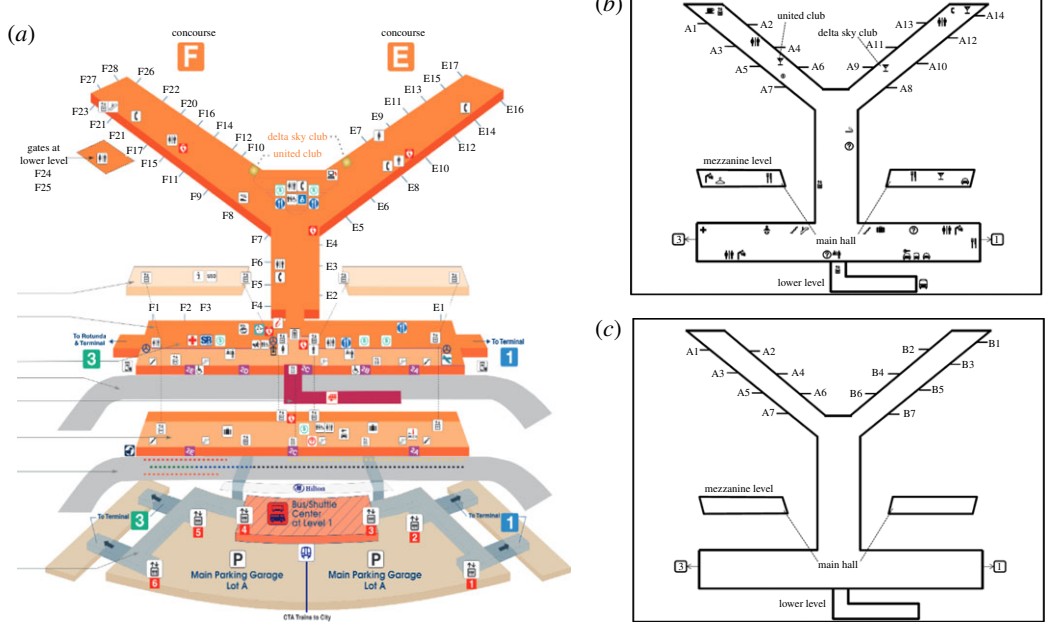

**Figure 1.** (*a*) O'Hare Airport Terminal 2 map. Source: http://airportczar.com/ohare/map/. (*b*) Complex map for the present studies. (*c*) Simple map, redesigned and simplified.

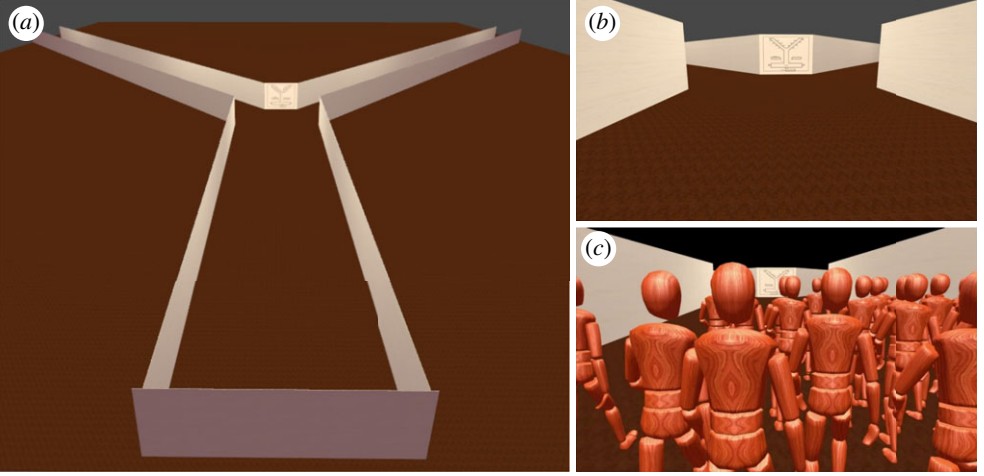

**Figure 2.** (*a*) Overview of the virtual environment used for all three studies. (*b*) View of the front wall of the intersection from an avatar's perspective during one of the videos in Study 1. (*c*) View of the other avatars from a first-person perspective during Study 2.

compensation. Before the start of each video, participants were instructed to choose whether to turn left or right to reach a selected gate (e.g. A2). After clicking the start button, the video started playing. While the video was playing, participants could choose to go left or right by clicking buttons on the computer screen. As the avatar moved closer to the map wall, participants gained a better view of the map, which should have made it easier to make a decision. They were allowed to change their decisions throughout each trial. Each participant was asked to respond to eight types of videos (i.e. simple/complex map designs by original/mirrored map orientation by left/right correct responses) for each of three trials. The three trials for each video type only varied with respect to the gate number of the goal. The order of these 24 trials was predetermined and randomized for each participant.

## 2.4. Design

The only independent variable of interest was map type (simple versus complex; within-subjects). The three dependent variables were the amount of time from the beginning of each video to each participant's final decision, the percentage of errors in participants' final decisions, and the number of mouse clicks recorded

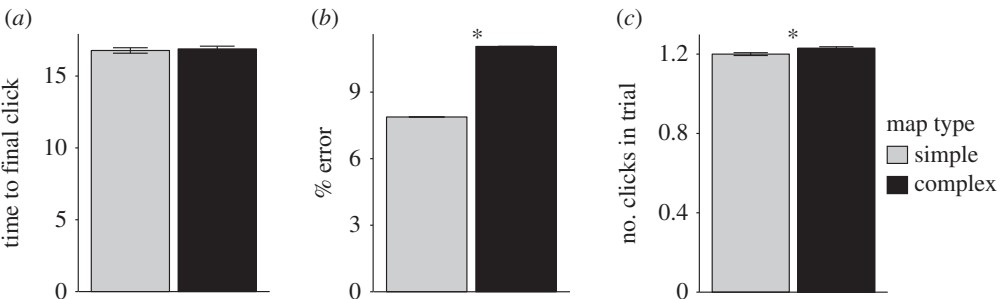

**Figure 3.** Results from Study 1. For all three graphs, the error bars represent standard error of the difference between means. (*a*) Difference between simple and complex maps in terms of time to final click ($p = 0.721$). (*b*) Significant difference between simple and complex maps in terms of percentage of errors in participants' final decisions ($p = 0.001$). (*c*) Significant difference between simple and complex maps in terms of number of clicks ($p = 0.049$). There are 145 data points in total represented by each bar. The asterisks '*' denote a significant effect.

during each trial. Two-tailed, paired-sample *t*-tests were used to analyse the effect of map type on these three dependent variables.

## 2.5. Results

Simple maps resulted in lower values (compared to complex maps) for all three dependent measures. Levene's test revealed no difference between the variances of the two groups in terms of time to final click ($p = 0.611$), percentage of errors ($p = 0.071$), or number of clicks ($p = 0.959$). The paired-sample *t*-tests revealed that the difference between simple and complex maps was not significant in terms of time to final decision, $t_{144} = 0.36$, s.e. $= 0.38$, $p = 0.721$, $d = 0.019$. However, we found significant differences between simple and complex maps in terms of percentage of errors in participants' final decisions, $t_{144} = 3.30$, s.e. $= 0.01$, $p = 0.001$, $d = 0.204$, and number of clicks during the trial, $t_{144} = 1.99$, s.e. $= 0.01$, $p = 0.049$, $d = 0.113$ (figure 3). In addition, there was a significant linear effect of trial on time to final decision, $F_{1,144} = 35.966$, m.s.e. $= 115.362$, $p < 0.001$, $d = 0.830$, suggesting that participants improved over trials.

# 3. Study 2

Study 1 provides evidence that map complexity can affect navigation decisions. In order to observe and collect crowd movement data, Study 2 was designed so that participants could control their own movement through the virtual environment amongst a crowd of other human-controlled avatars.

## 3.1. Participants

Twenty-eight participants (16 men and 12 women) were recruited via the University Registration Center for Participants (www.uast.uzh.ch). All of these participants were between 20 and 29 years of age (mean age $= 24.3$). Each participant was paid between 25 CHF and 30 CHF, depending on their performance.

## 3.2. Materials

For Study 2, we used the same Y-shaped virtual environment from Study 1. Participants had the first-person perspective of the avatar and were able to move through the virtual environment by using the arrow or the 'WASD' keys (e.g. the up arrow or 'W' for forward movement) on the keyboard. The maximum forward movement speed was $1.3 \text{ m s}^{-1}$, and the maximum backwards and sideways movement speeds were $0.6 \text{ m s}^{-1}$. Participants could also use the mouse to rotate their field of view up to a maximum angular velocity of $120° \text{ s}^{-1}$. Participants could see each of the other participants' avatars that were within their field of view. These avatars were represented by wooden mannequins with an eye height of 1.8 m and a collision radius of 0.25 m. Also, the design of the simple/complex maps and the selection of target gates were the same as in Study 1.

The experiment was conducted in the Decision Science Laboratory (DeSciL) at ETH Zürich. The DeSciL is a laboratory that consists of 36 cubicles, each of which is equipped with a desktop

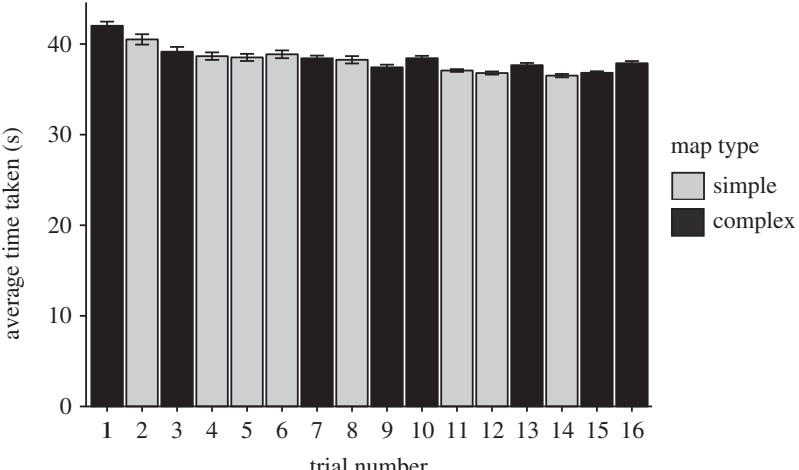

**Figure 4.** Mean time required to finish each trial for Study 2.

computer. This networked computer set-up has been used previously for studies on group decision-making in both game theory [56] and crowd dynamics [9] contexts. The laboratory and experiment set-up were explained in detail by Zhao *et al.* [46]. Each participant performed the experiment on a Dell Optiplex 980 computer running Windows 7 Enterprise SP1 X64 and connected to 19-inch diagonal Dell 1909W monitors with a resolution of $1920 \times 1080$ pixels. The application frame rate was at least 30 frames per second, and network latency was approximately 67 ms.

## 3.3. Procedure

After reading and signing an informed consent form, participants were shown how to use the mouse-and-keyboard control interface using an interactive tutorial [9,42]. For each trial, participants were given a gate number and were asked to move down a virtual corridor towards a Y-shaped intersection. Their task was to turn left or right at this intersection in order to reach the given gate. All of the participants were located in the same virtual corridor at the same time, and the other participants' avatars were visible. However, participants were not instructed as to whether the other participants were given the same gate as a goal (even though they were given the same gate within one trial). For the training trial at the beginning of the experiment, no map was displayed in order to familiarize participants with the environment and control interface. During this training trial, the participants were simply asked to move to the left at the intersection.

For 16 subsequent trials, the map randomly varied between simple and complex designs, and each avatar's starting location was randomly selected from a grid of 36 locations. During each trial, the virtual reality system automatically recorded the time and the coordinates of all participants' avatars between 20 and 60 times per second (depending on Unity's update rate).

## 3.4. Design

As in Study 1, the only independent variable of interest was map type (simple versus complex; within-subjects). The two dependent variables were the percentage of errors and time required to reach the end of the corridor. We analysed the effect of map type on each dependent variable using two-tailed, paired-sample *t*-test.

## 3.5. Results

Only one error was committed throughout the entire experiment, so the analyses were focused on the time required to reach the end of the corridor in the two map conditions. The corresponding *t*-test revealed that trials with complex maps (mean = 38.48 s) did not require significantly more time to finish than trials with simple maps (mean = 38.15 s, figure 4), $t_{26} = -1.29$, s.e. = 0.253, $p = 0.208$, $d = 0.131$. In addition, we found a significant linear effect of trial on time required to complete each trial,

$F_{1,28} = 16.31$, m.s.e. $= 36.37$, $p < 0.001$, $d = 1.969$. We speculate that this effect may be attributable to familiarity with the control interface in VR [57,58].

# 4. Study 3

In Study 3, we replayed the trajectories of participants from Study 2 and, for some trials, reversed the direction of crowd movement. This manipulation caused the direction indicated by crowd movement to conflict with the direction indicated by the map. Participants moved through the same virtual environment individually among these replayed trajectories.

## 4.1. Participants

Twenty-nine additional participants (20 men and 9 women) were recruited via the University Registration Center for Participants (http://www.uast.uzh.ch) and paid 30 CHF upon completion of the experiment. All participants were between the ages of 18 and 27 (mean age = 22.38).

## 4.2. Materials

The same virtual environment and physical apparatuses from Study 2 were used in Study 3. However, participants performed Study 3 individually amongst computer-simulated agents. These agents' trajectories were equivalent to those recorded from the participants of Study 2. The participants were told that the crowd movements were from real people. We excluded the trajectories of the participants from Study 2 who made incorrect choices. In order to avoid collisions between the individual participants' avatars and the computer-simulated agents, the participants' initial position was behind all of the agents (at least 3 m). Four different types of trials were designed to include the combination of two map types (simple versus complex) and two different crowd movements (original versus reversed). 'Original' crowd movement refers to a crowd that moved in the same direction as indicated by the map, and 'reversed' crowd movement refers to a crowd that moved in the opposite direction. To simulate a reversed crowd, we flipped the indicated destination. For example, if gate A2 was indicated as the correct destination in one trial in Study 2, B2 could be used as the correct destination in the corresponding trial of Study 3. In this example, the crowd would still move towards A2.

## 4.3. Procedure

The procedure for Study 3 was similar to that for Study 2, except for the order of the trials and the addition of trials in which the direction indicated by the crowd was different from the direction indicated by the maps. Participants underwent the same training procedure as Study 2 before completing 32 testing trials. For the order and types of the trials, see figure 5.

In order to ensure that participants would consider crowd movement as a reliable cue, the first eight trials only contained the original crowd without the conflict between cues. If the conflict between crowd movement and the map appeared earlier, then participants would probably deem crowd movement as an unreliable cue. The remaining trials (9–32) were in the same random and predetermined order for all participants.

## 4.4. Design

The two within-subjects independent variables were map type (simple versus complex) and crowd movement (original versus reversed). The dependent variables were the number of errors (with respect to the direction indicated by the map), the time taken to finish the trial, the number of hesitation points within the entire $y$-shaped corridor, and the number of hesitation points within the area from which the map was visible. According to Filippidis and colleagues [59], such an area can be defined as the visible catchment area (VCA). We first compared trial 9 (i.e. the first trial in which participants faced reversed crowd movement) to all of the preceding trials and all of the subsequent trials in terms of number of errors using two separate one-proportion $Z$-tests. For each of the four dependent variables, we then conducted separate two (map type) by two (crowd movement) analyses of variance (ANOVAs). When Levene's tests revealed heterogeneity of variance among the experimental conditions, we also performed an aligned rank transform analysis of variance (ART ANOVA) [60]. The ART ANOVA is a

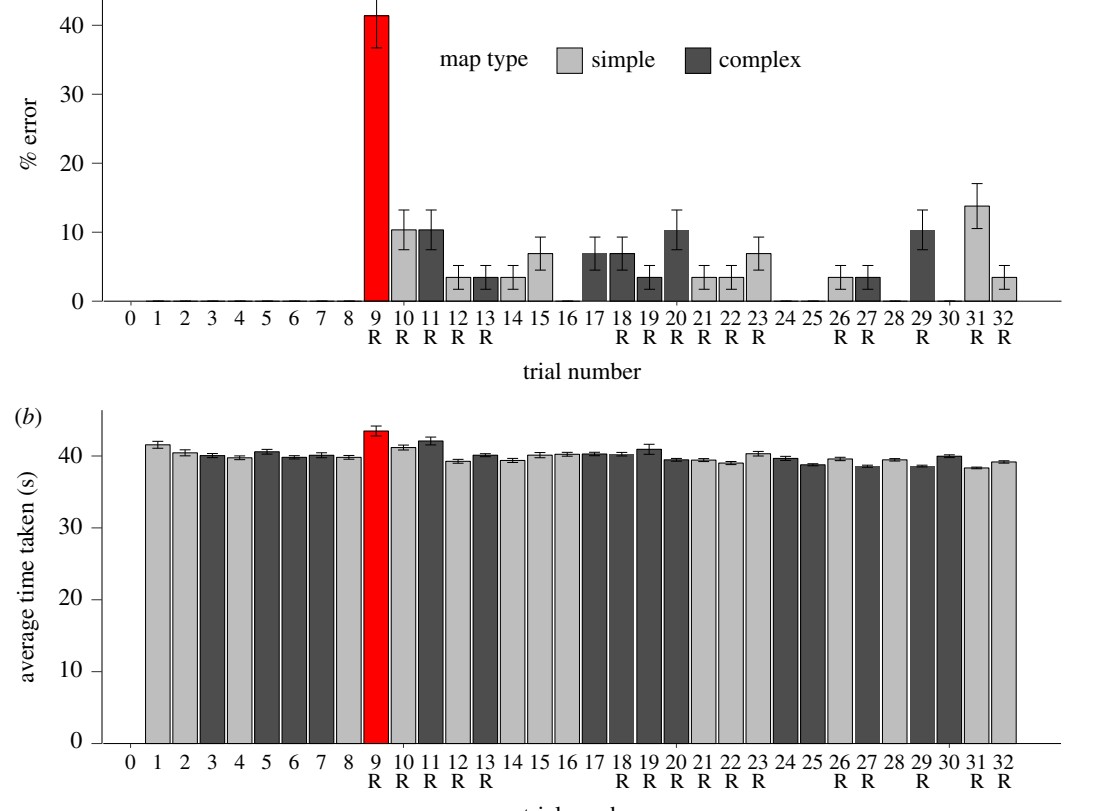

**Figure 5.** (*a*) Mean number of errors for each trial in Study 3. (*b*) Mean completion time for each trial in Study 3. For some trials, the original crowd from Study 2 was employed (e.g. trials 1 to 8), but for reversed trials (e.g. trials 9–13), the crowd indicated a different direction to the map. The red highlighted trial 9 is the first trial in which the crowd trajectories were reversed. The trials with reversed crowd movement are annotated with an 'R' below the trial number. The trials without an 'R' represented the original crowd movement.

non-parametric version of the typical ANOVA that computes each main effect and interaction by first aligning the data with respect to a specific effect and then converting the data to ranks. Hesitation points were defined as successive data points within the same $m^2$ and within a time window of 0.13 s (i.e. 10% less than the mean amount of time between two successive data points within the same $m^2$). The VCA was defined as a set of locations from which the map could have been visible [59]. For our purposes, this value (15.58 m from the front of the map) was derived from Study 1 by inferring the average location at which the final decisions were made. Specifically, we considered the radius of the VCA as the mean distance from the map where participants made their final click in Study 1, indicating their final decision. Kernel density estimates (KDE) were then used to compare the distributions of hesitation points within the VCA between pairs of experimental conditions [61]. KDE is a multivariate kernel discriminant analysis that compares two distributions of data [61,62]. In order to simplify the KDE analyses, locations within the VCA were excluded if they did not contain any hesitation points.

## 4.5. Results

The number of errors for trial 9 was significantly higher than the eight preceding trials, $Z = 4.52$, $p < 0.001$, and all of the subsequent trials (in aggregate), $z = 3.99$, $p < 0.001$ (figure 5*a*). In addition, trial 9 tended to require more time (43.48 s) than all of the other trials (mean = 40.12, figure 5*b*). The results of the $2 \times 2$ ANOVAs for number of errors, completion times, number of hesitation points and number of hesitation points within the VCA are presented in table 1. Because of heterogeneity of variance, number of errors was also analysed using an ART ANOVA (table 2). Both parametric and non-parametric analyses revealed a main effect of crowd movement on number of errors, whereas the effect of map types on number of errors was only revealed by the parametric analysis. In addition,

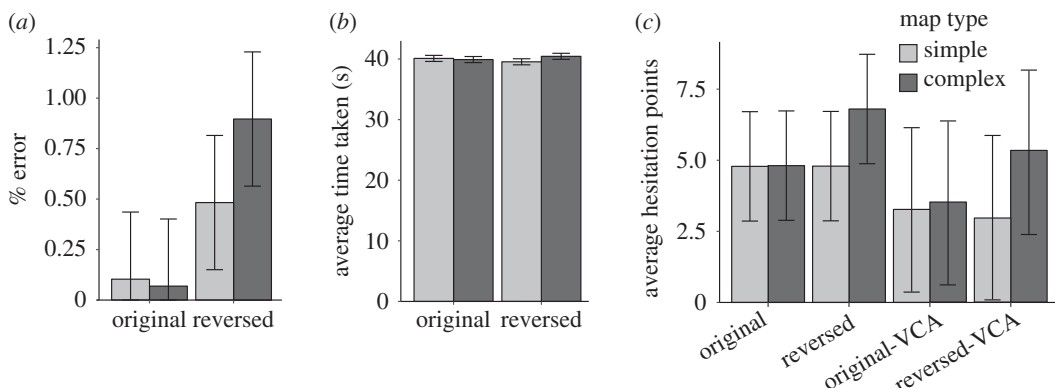

**Figure 6.** (*a*) Mean per cent error for each experimental condition. (*b*) Mean completion times for each experimental condition. (*c*) Mean number of hesitation points for the full environment and inside the VCA. Error bars represent standard error. Original and reversed represent original and reversed crowd movement, respectively.

**Table 1.** Two-way ANOVA results for all four dependent variables from Study 3. The asterisks denote significant effects.

| measure | contrast | F | m.s.e. | p |
|---|---|---|---|---|
| number of errors | 2 × 2 interaction | 10.755 | 0.135 | 0.003* |
| | effect of crowd movement | 4.717 | 2.239 | 0.038* |
| | effect of map | 9.108 | 0.115 | 0.005* |
| time | 2 × 2 interaction | 8.591 | 1.003 | 0.007* |
| | effect of crowd movement | 0.003 | 2.056 | 0.958 |
| | effect of map | 2.189 | 1.680 | 0.150 |
| number of | 2 × 2 interaction | 1.926 | 14.799 | 0.176 |
| hesitation points | effect of crowd movement | 1.670 | 17.364 | 0.207 |
| | effect of map | 1.021 | 29.396 | 0.321 |
| number of | 2 × 2 interaction | 1.919 | 33.004 | 0.177 |
| hesitation points | effect of crowd movement | 1.815 | 16.406 | 0.189 |
| inside VCA | effect of map | 2.620 | 49.957 | 0.117 |

**Table 2.** Two-way ART ANOVA results for number of errors from Study 3. The asterisk denotes a significant effect.

| | d.f. | Df.res | F | p |
|---|---|---|---|---|
| effect of map | 1 | 28 | 1.3633 | 0.25 |
| effect of crowd movement | 1 | 28 | 34.9435 | <0.001* |
| 2 × 2 interaction | 1 | 28 | 2.5140 | 0.12 |

there was a significant interaction between crowd movement and map complexity in terms of time ($p = 0.007$) and number of errors ($p = 0.003$). Per cent error (figure 6*a*) and the mean number of hesitation points (figure 6*c*) were both highest for trials with reversed crowd movement and complex maps. Time required per trial was similar for all four types of trials (figure 6*b*).

In order to visualize hesitation points within the VCA, we created 'normalized' density maps to weigh each hesitation point according to its (temporal) length, and higher values represent more hesitation in general (figure 7). Each cell is of the size of $1 \times 1$ m because the hesitation point is defined with the same size scale. From these two sets of density maps, the complex/reversed condition clearly exhibits the most hesitation. Accordingly, the KDE analyses for normalized density maps revealed a significant difference

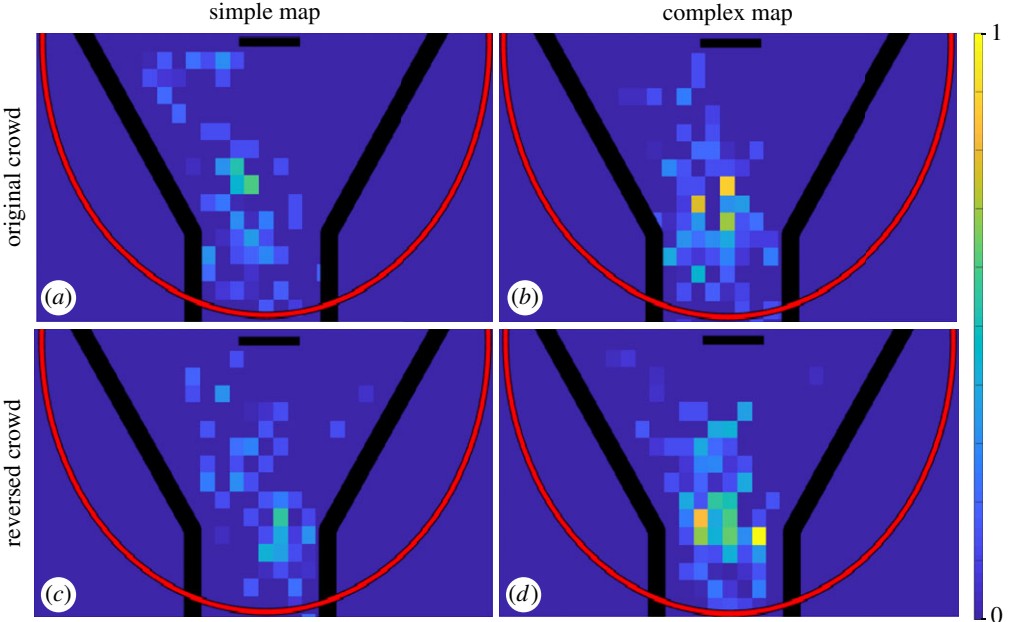

**Figure 7.** Normalized density maps of hesitation points. The red semi-circle represents the boundary of the visible catchment area. The scale ranged from 0 (no hesitation points) to the maximum number of hesitation points for any location across the four experimental conditions. For all density maps, we combined trials with goal locations to the left and right. (*a*) Simple maps and the original crowd movement. (*b*) Complex maps and the original crowd movement. (*c*) Simple maps and reversed crowd movement. (*d*) Complex maps and reversed crowd movement. There is a significant difference between simple/reversed and complex/reversed conditions ($p = 0.02$). The size of each cell is $1 \times 1$ m.

between simple/reversed and complex/reversed conditions ($p = 0.02$). However, this effect was not confirmed using the aggregated density maps ($p = 0.14$). All of the other comparisons were not significant, including simple/original versus simple/reversed (aggregated $p = 0.51$, normalized $p = 0.57$), simple/original versus complex/original (aggregated $p = 0.53$, normalized $p = 0.39$) and complex/original versus complex/reversed (aggregated $p = 0.50$, normalized $p = 0.51$).

## 5. Discussion

In the present studies, we investigated the effects of map design and crowd movement on spatial decision-making in VR. First, we used an online crowd-sourcing platform to present videos of movement through a Y-shaped intersection from a ground-level perspective and asked participants to decide whether to turn left or right. Study 1 revealed that a complex map design led to longer decision times and more mistakes compared to a simpler map design. In the next study, we collected participants' trajectories as they moved together through the same Y-shaped intersection using a networked VR laboratory. Study 2 revealed a very low number of errors and slower decisions with complex map design than with the simple map design, but the results were not significant. Study 2 also allowed us to collect crowd trajectories for the final study. In the third study, we flipped the indicated destination to produce a conflict between the direction indicated by the map and the direction indicated by the crowd. The results from Study 3 demonstrated a significant effect of conflicting crowd movement on the number of incorrect decisions. Study 3 also showed that a complex map design can lead to a higher number of incorrect decisions and a different distribution of hesitation points in the environment when in conflict with crowd movement.

Together, these three studies demonstrated the expected differences between simple and complex maps. Consistent with O'Neill [27], we found that simple maps may help people make more accurate spatial decisions. However, these findings may appear to conflict with previous research that did not find an effect of different map designs on decision-making [35,36]. While Soh & Smith-Jackson [35] varied other aspects of map design, Devlin & Bernstein [36] varied the amount of detail in the maps as in the present studies. In Devlin & Bernstein [36], participants navigated through a complex environment with several intersections rather than a simple one-intersection environment. This complex environment may have

allowed participants more time and space to make a decision or correct for a previous decision [36]. In addition, the less-detailed maps from Devlin & Bernstein [36] were still relatively more complex than our simple maps in terms of both the absolute number of details and the number of details per unit of space. Because the differences between maps in the present study were more pronounced, the effect of map design on decision time and accuracy might have been easier to detect.

We did not find an effect of map design on decision accuracy in Study 2, but this result is due to the very low number of incorrect decisions overall. This low number of incorrect decisions in Study 2 may be attributable to the co-presence of participants in the same virtual environment. This co-presence allowed participants to 'physically' interact with each other by reacting to each other's visually depicted movements in real time. Moussaïd *et al.* [9] found that following behaviour could lead to incorrect collective decisions during navigation in VR. In the present study, these interactions between participants may have led to more accurate decisions via collective intelligence [14], although both studies were conducted in the same laboratory. Notably, participants in Study 1 were alone in the environment, and participants in Study 3 were immersed with computer-controlled avatars.

Despite the avatars in Study 3 being controlled by the computer, participants sometimes followed the virtual crowd as if it were controlled by other participants. Indeed, the conflict between the direction indicated by the map and the direction indicated by the virtual crowd led to significantly more incorrect decisions (with respect to the map). Although we only tested for following behaviour using one group of participants, this finding is consistent with previous research that found that participants tended to hesitate before responding to a similar conflict between the direction indicated by a computer-controlled avatar and the actual direction of an emergency exit [6]. Both findings indicate that the computer-controlled avatars were somewhat believable, despite lack of direct communication between the participants and avatars. In the present study, these avatars also lacked visual communication cues such as eye contact and hand gestures. While these features may be added in future studies to improve believability, the social signal produced by our relatively simple avatars was sufficient to elicit following behaviour.

We also observed an interaction between the direction of crowd movement and map design for both the time required to complete a trial and the distribution of hesitation points. In both cases, there was only a difference between simple and complex maps when the direction of crowd movement was reversed. We interpret this interaction as an indication that the conflict trials resulted in higher cognitive load than trials without this conflict. Thus, these findings support the notion that map designers should simplify maps by reducing the number of extraneous icons, especially in cases of high cognitive load [28]. In addition, maps may be personalized by only including icons that are relevant for each person with a specific task. The convenience and ubiquity of smartphones allows for maps in airports to be displayed with an interactive digital device on which information regarding other types of locations can be presented for different tasks (e.g. searching for the nearest bathroom, finding a restaurant). The findings from our study can inspire such map designs to be simplified and more task-specific for each individual user.

Another important finding was the difference in the distribution of hesitation points caused by the more complex map design and crowd movement. Previous research has already shown that hesitations by pedestrians can increase the probability of collisions and decrease the moving speed of a crowd [63,64]. Such involuntary slowing down of crowds can create unexpected obstacles and lead to congestion [65]. This finding reinforces the idea that the task-specific and individualized presentation of map information may influence hesitation.

One issue that has not been addressed in the present study is whether the effects of crowd movement and map complexity would apply to more complicated wayfinding tasks that require more than just binary choices from the participants. Future research can complement the present studies by introducing manipulations of time pressure, task difficulty and visual noise. Time pressure can be introduced in order to simulate the stress associated with finding a terminal gate. We would expect time pressure to increase the size of the effect of map complexity. This idea is consistent with previous research [9], which has found that people are much more likely to follow others in an evacuation scenario with additional time pressure. In addition, adding visual noise (e.g. smoke from a simulated fire) would probably reduce the size of the effect of map complexity and increase following behaviour. Following vision-based models of collective motion [4,5,66], we would expect, especially with low visibility, for objects closer to the navigator to affect the navigator's behaviour more than objects further from the navigator. Adding noise to the virtual environment (e.g. additional signs on the ceiling and walls) could decrease the observed effect of map complexity, especially if these signs are irrelevant to the task. Finally, looking at the interaction between task difficulty and map

complexity would be interesting because we could expect the effect of map complexity to either decrease or increase with task difficulty. Task difficulty could be increased by having participants find the actual gate instead of turning left or right. On the one hand, the effect of map complexity could decrease with task difficulty because the task would be longer and involve more decisions along the route to the destination. On the other hand, the effect of map complexity could increase with task difficulty because the time required to find the destination is amplified by each decision along the route.

Another possible limitation of the present study is participants' habituation to the VR task over trials, which may help explain some of the main effects for map type and crowd movement that we originally obtained. Unfortunately, for our experimental design, we could not test for a higher order interaction between map type, crowd movement, and trial number. For a future study, we could include additional trials after trial 9 and exclude trials before this probe trial in order to maintain a balance among the various experimental conditions. Prior studies have revealed the importance of map design [27,35,36,67–69] and crowds [6,9–11,70] as wayfinding cues, but to our knowledge, the present studies represent the first investigations of the interaction between these two factors. These studies also contribute to the literature by demonstrating that map design can affect human decision-making in VR and that crowd movement can affect spatial behaviour, even when the individual avatars are computer-controlled. Both of these contributions may inform future studies on collective intelligence, especially in situations with high cognitive load. For example, maps may need to be designed more simply in public spaces with larger crowds. In the future, a similar approach can be used for the preoccupancy evaluation of indoor environments.

Ethics. All of these methods were approved by the Ethics Commission of ETH Zürich (EK 2015-N-37). Participants have read and signed consent in all experiments.

Data accessibility. The experiment data are available in the electronic supplementary material.

Authors' contributions. H.Z., T.T., M.K., M.M., C.H., and V.R.S. designed research. H.Z. and T.T. performed the experiments. H.Z., T.T. and A.G, analysed data and wrote the paper. H.Z. and T.T. contributed equally to this work. All authors reviewed the manuscript and gave final approval for publication.

Funding. The study was funded by the Swiss National Science Foundation as part of the grant 'Wayfinding in Social Environments' (grant no. 100014-162428). M.K. was funded in part by NSF (grant nos. IIS-1703883 and S&AS-1723869).

Acknowledgements. We thank Stefan Wehrli, Sephora Madjiheurem and Alvaro Ingold for their support during experiment implementation and data collection. Preliminary results from Study 2 were previously used as representative experiment in the paper by Zhao et al. [46].

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
