## [Reviewer comments · Royal Society Open Science]

Review History

RSOS-191523.R0 (Original submission)

Review form: Reviewer 1

Is the manuscript scientifically sound in its present form?

Yes

Are the interpretations and conclusions justified by the results?

Yes

Is the language acceptable?

Yes

Do you have any ethical concerns with this paper?

No

Have you any concerns about statistical analyses in this paper?

No

Recommendation?

Major revision is needed (please make suggestions in comments)

Comments to the Author(s)

The authors present three studies in which participants were presented with a Y shaped corridor. The corridor illustrated an airport terminal, and in each study participants were tasked to find a gate indicated on a map. The authors manipulated the complexity of a terminal map (study 1-3), and then introduced simulated crowd behavior. In study 1, participants in an online platform were recruited; In study 2 and 3, participants were tested in a desktop VR environment. The authors found that more complex maps, increased the number of mistakes participants made when they had to decide where to go at the intersection.

I enjoyed reading this manuscript. The methods section of the study is very solid. The authors communicate their work very clearly and the sequence of studies make intuitively sense. Most of my comments are pretty general and more curiosity driven. Many of my points could either be addressed in the discussion section, by collecting more data, or be ignored.

Overall I am not terribly surprised by the results (in particularly study 1 and 2); given the very low error rate in both scenarios, I am wondering if the task itself was too easy. The simplified map essentially reduced the noise for the specific task at hand (e.g., when I need to find my gate, I do not need to know where the restrooms are). However, I wonder if the effects found here would hold up in a more difficult task (i.e., one that produces more errors at baseline). The authors could, for instance, task participants with finding an arbitrary item on the map, add time pressure (e.g., by using an evacuation scenario) or add visual noise to the virtual environment.

I am a little worried about the ecological validity of the virtual environment. I appreciate that the authors created a very reduced and controlled virtual environment; however, adding a little more complexity to the virtual environment so that it is closer to an actual terminal would probably be helpful. For instance, signage (which could be a confounding factor) is completely missing in the virtual environments. The authors could consider, recreating experiment 1 in a photomontage of O'Hare.

Even the complex map that the authors presented appears simpler than the actual 3D terminal map shown Fig 1a. Did the authors consider comparing 3D maps to 2D maps?

Would it be realistic to expect generic airport maps to be simplified? The reason for their complexity, I assume, is to provide a flexible tool so that users can find a number of different locations (e.g., restaurants, terminals and restrooms). What would be the real life implications of the findings? Would the authors suggest developing task specific maps (e.g., in a smartphone application)?

For methodological reasons, the authors placed the participants behind the crowd in experiment 3. However, when navigating airports, we are rarely following a crowd; that is, the signal from the social influence is probably noisier than in the studies presented here. I believe that there is also some research that suggest that crowd density (i.e., the number and distance to neighbors in a crowd) influences the how strongly people are influenced by a crowd. A quick google scholar search only revealed the reference below, but there might be more out there that the authors want to discuss:

T. D. Wirth and W. H. Warren, "The visual neighborhood in human crowds: Metric vs. Topological Hypotheses," *Journal of Vision*, vol. 16, pp. 982-982, 2016.

I really liked the normalized hesitation heat maps. Did the authors consider plotting difference maps? This might make a comparison between the conditions more intuitive. I am not sure about this though.

The authors probably played around with the cell sizes of the density maps. These should be reported in the figure caption; and also a rationale for the specific cell size should be given.

I was wondering about the consequences of identifying hesitation points. These could create unexpected obstacles and impede crowd flow in general.

Did the authors consider analyzing the heading data that could be derived from the mouse movement?

Review form: Reviewer 2

Is the manuscript scientifically sound in its present form?

Yes

Are the interpretations and conclusions justified by the results?

Yes

Is the language acceptable?

Yes

Do you have any ethical concerns with this paper?

Yes

Have you any concerns about statistical analyses in this paper?

Yes

Recommendation?

Accept with minor revision (please list in comments)

Comments to the Author(s)

Overall, this is an interesting study. I particularly like how the authors have tested their hypotheses in a series of studies rather than only performing one experiment. In the context of studying the effect of map complexity, it would have been nice to see a wayfinding experiment that goes beyond choosing between only 2 options.

I have some comments that should be easy to address by re-writing parts of the text. Please take the time to do so.

Kind regards.

Comments

Just a note on page 2: "...likely to herd as a crowd...": it has recently been suggested that several terms, including "herding" are not sufficiently precise. Seeing that you refer to "following behaviour" in the same sentence, why not just use the latter, more precise, phrase. See: Haghani, M., Cristiani, E., Bode, N. W., Boltes, M., & Corbetta, A. (2019). Panic, irrationality, and herding: three ambiguous terms in crowd dynamics research. *Journal of advanced transportation*, 2019.

Page 2: "For example, Moussaïd and colleagues [7] validated a multi-user VR framework using a study of collective navigation behaviour and herding during a stressful evacuation."

The comparisons of the VR framework in [7] focussed on operational level behaviour (passing another person in a corridor and flow depending on bottleneck width), aspects that are dependent on simple physics in the VR (walking speed, size of avatars, etc). The match between VR and experiments was not exactly perfect. So what do you mean by "validated" here?

Study 1: each participant performed 24 trials. You should test for habituation effects in your statistical analysis. In the caption on figure 3, it would be useful to indicate the number of data points making up the bar charts displayed (145x12?). Did you check the t test assumptions hold? The size of the difference in the number of clicks in fig. 3 is very small. It would be good to mention this.

Study 2: again, it would be good to explicitly mention the small effect size in the difference in time taken to complete trials. Also, please discuss the habituation effect (reduction in time taken with trial number) which seems to be visible in fig. 4. Finally, it wasn't clear to me if participants all had the same goal?

Study 2: you only use one group of participants. This is a major limitation of your study as it is not clear how your results would vary if another group of participants performed the experiment. You must state this limitation explicitly somewhere in the text.

Captions of table 1 and 2: please indicate which study the results presented in the table are from.

Decision letter (RSOS-191523.R0)

22-Nov-2019

Dear Mr Zhao,

The editors assigned to your paper ("The interaction between map complexity and crowd movement on navigation decisions in virtual reality") have now received comments from reviewers. We would like you to revise your paper in accordance with the referee and Associate Editor suggestions which can be found below (not including confidential reports to the Editor). Please note this decision does not guarantee eventual acceptance.

Please submit a copy of your revised paper before 15-Dec-2019. Please note that the revision deadline will expire at 00.00am on this date. If we do not hear from you within this time then it will be assumed that the paper has been withdrawn. In exceptional circumstances, extensions may be possible if agreed with the Editorial Office in advance. We do not allow multiple rounds of revision so we urge you to make every effort to fully address all of the comments at this stage. If deemed necessary by the Editors, your manuscript will be sent back to one or more of the original reviewers for assessment. If the original reviewers are not available, we may invite new reviewers.

- Data accessibility

<http://datadryad.org/submit?journalID=RSOS&manu=RSOS-191523>

- Competing interests

- Authors' contributions

- Acknowledgements

- Funding statement

Kind regards,
Lianne Parkhouse

Editorial Coordinator
Royal Society Open Science
openscience@royalsociety.org

on behalf of Dr Narayanan Srinivasan (Associate Editor) and Essi Viding (Subject Editor)
openscience@royalsociety.org

Associate Editor's comments (Dr Narayanan Srinivasan):

Two reviewers have now commented on the paper. The authors are requested to address all the comments point by point and revise the paper accordingly.

Reviewers' Comments to Author:

Reviewer: 1

Comments to the Author(s)

The authors present three studies in which participants were presented with a Y shaped corridor. The corridor illustrated an airport terminal, and in each study participants were tasked to find a gate indicated on a map. The authors manipulated the complexity of a terminal map (study 1-3), and then introduced simulated crowd behavior. In study 1, participants in an online platform were recruited; In study 2 and 3, participants were tested in a desktop VR environment. The authors found that more complex maps, increased the number of mistakes participants made when they had to decide where to go at the intersection.

I enjoyed reading this manuscript. The methods section of the study is very solid. The authors communicate their work very clearly and the sequence of studies make intuitively sense. Most of my comments are pretty general and more curiosity driven. Many of my points could either be addressed in the discussion section, by collecting more data, or be ignored.

Overall I am not terribly surprised by the results (in particularly study 1 and 2); given the very low error rate in both scenarios, I am wondering if the task itself was too easy. The simplified map essentially reduced the noise for the specific task at hand (e.g., when I need to find my gate, I do not need to know where the restrooms are). However, I wonder if the effects found here would hold up in a more difficult task (i.e., one that produces more errors at baseline). The authors could, for instance, task participants with finding an arbitrary item on the map, add time pressure (e.g., by using an evacuation scenario) or add visual noise to the virtual environment.

I am a little worried about the ecological validity of the virtual environment. I appreciate that the authors created a very reduced and controlled virtual environment; however, adding a little more complexity to the virtual environment so that it is closer to an actual terminal would probably be helpful. For instance, signage (which could be a confounding factor) is completely missing in the virtual environments. The authors could consider, recreating experiment 1 in a photomontage of O'Hare.

Even the complex map that the authors presented appears simpler than the actual 3D terminal map shown Fig 1a. Did the authors consider comparing 3D maps to 2D maps?

Would it be realistic to expect generic airport maps to be simplified? The reason for their complexity, I assume, is to provide a flexible tool so that users can find a number of different locations (e.g., restaurants, terminals and restrooms). What would be the real life implications of the findings? Would the authors suggest developing task specific maps (e.g., in a smartphone application)?

For methodological reasons, the authors placed the participants behind the crowd in experiment 3. However, when navigating airports, we are rarely following a crowd; that is, the signal from

the social influence is probably noisier than in the studies presented here. I believe that there is also some research that suggest that crowd density (i.e., the number and distance to neighbors in a crowd) influences the how strongly people are influenced by a crowd. A quick google scholar search only revealed the reference below, but there might be more out there that the authors want to discuss:

T. D. Wirth and W. H. Warren, "The visual neighborhood in human crowds: Metric vs. Topological Hypotheses," *Journal of Vision*, vol. 16, pp. 982-982, 2016.

I really liked the normalized hesitation heat maps. Did the authors consider plotting difference maps? This might make a comparison between the conditions more intuitive. I am not sure about this though.

The authors probably played around with the cell sizes of the density maps. These should be reported in the figure caption; and also a rationale for the specific cell size should be given.

I was wondering about the consequences of identifying hesitation points. These could create unexpected obstacles and impede crowd flow in general.

Did the authors consider analyzing the heading data that could be derived from the mouse movement?

Reviewer: 2

Comments to the Author(s)

Overall, this is an interesting study. I particularly like how the authors have tested their hypotheses in a series of studies rather than only performing one experiment. In the context of studying the effect of map complexity, it would have been nice to see a wayfinding experiment that goes beyond choosing between only 2 options.

I have some comments that should be easy to address by re-writing parts of the text. Please take the time to do so.

Kind regards.

Comments

Just a note on page 2: "...likely to herd as a crowd...": it has recently been suggested that several terms, including "herding" are not sufficiently precise. Seeing that you refer to "following behaviour" in the same sentence, why not just use the latter, more precise, phrase. See: Haghani, M., Cristiani, E., Bode, N. W., Boltes, M., & Corbetta, A. (2019). Panic, irrationality, and herding: three ambiguous terms in crowd dynamics research. *Journal of advanced transportation*, 2019.

Page 2: "For example, Moussaïd and colleagues [7] validated a multi-user VR framework using a study of collective navigation behaviour and herding during a stressful evacuation."

The comparisons of the VR framework in [7] focussed on operational level behaviour (passing another person in a corridor and flow depending on bottleneck width), aspects that are dependent on simple physics in the VR (walking speed, size of avatars, etc). The match between VR and experiments was not exactly perfect. So what do you mean by "validated" here?

Study 1: each participant performed 24 trials. You should test for habituation effects in your statistical analysis. In the caption on figure 3, it would be useful to indicate the number of data points making up the bar charts displayed (145x12?). Did you check the t test assumptions hold? The size of the difference in the number of clicks in fig. 3 is very small. It would be good to mention this.

Study 2: again, it would be good to explicitly mention the small effect size in the difference in

time taken to complete trials. Also, please discuss the habituation effect (reduction in time taken with trial number) which seems to be visible in fig. 4. Finally, it wasn't clear to me if participants all had the same goal?

Study 2: you only use one group of participants. This is a major limitation of your study as it is not clear how your results would vary if another group of participants performed the experiment. You must state this limitation explicitly somewhere in the text.

Captions of table 1 and 2: please indicate which study the results presented in the table are from.

Author's Response to Decision Letter for (RSOS-191523.R0)

See Appendix A.

RSOS-191523.R1 (Revision)

Review form: Reviewer 1

Is the manuscript scientifically sound in its present form?

Yes

Are the interpretations and conclusions justified by the results?

Yes

Is the language acceptable?

Yes

Do you have any ethical concerns with this paper?

No

Have you any concerns about statistical analyses in this paper?

No

Recommendation?

Accept as is

Comments to the Author(s)

I thank the authors for responding to all of my comments; I think that this is really interesting work. Keep it up!

Review form: Reviewer 2

Is the manuscript scientifically sound in its present form?

Yes

Are the interpretations and conclusions justified by the results?

Yes

Is the language acceptable?

Yes

Do you have any ethical concerns with this paper?

No

Have you any concerns about statistical analyses in this paper?

Yes

Recommendation?

Accept with minor revision (please list in comments)

Comments to the Author(s)

Overall, I think the authors have dealt with my comments reasonably well. There are just a few outstanding issues that need to be addressed before this work can be published.

Kind regards.

****Comments****

For the habituation effects, please indicate the size of this effect, so that it can be compared to the other effects you report (e.g. for study 1, there is no data available to make this comparison). Also, I am not sure the randomised order of treatments ensures that habituation does not affect your findings, seeing that one random ordering is used in each study (e.g. in study 2 a new random order could have been created for each participant). A generalised linear model with predictors map type, trial number and possibly reversed crowd direction (study 3) and interaction terms would allow you to test this.

Line 32-33 on page 7: "...but this tendency is probably attributable to familiarity with the control interface in VR." Do you have any evidence to back this speculative statement up?

Comparing the results in tables 1 and 2, it seems that when using the ART ANOVA, you do not find a significant effect of map type on the number of errors.

Study 3: is it possible that the difference in the number of errors you find for the crowd movement treatment is entirely due to the outlier of trial number 9?

Page 11, lines 57-58: "Consistent with O'Neill [27], we found that simple maps helped people make faster [...] spatial decisions" given the small effect sizes and the non-significant differences in decision times, I don't think you can make this claim.

Decision letter (RSOS-191523.R1)

28-Jan-2020

Dear Mr Zhao:

On behalf of the Editors, I am pleased to inform you that your Manuscript RSOS-191523.R1 entitled "The interaction between map complexity and crowd movement on navigation decisions in virtual reality" has been accepted for publication in Royal Society Open Science subject to minor revision in accordance with the referee suggestions. Please find the referees' comments at the end of this email.

The reviewers and Subject Editor have recommended publication, but also suggest some minor revisions to your manuscript. Therefore, I invite you to respond to the comments and revise your manuscript.

- Ethics statement

- Data accessibility

<http://datadryad.org/submit?journalID=RSOS&manu=RSOS-191523.R1>

- Competing interests

- Authors' contributions

- Acknowledgements

- Funding statement

Please note that we cannot publish your manuscript without these end statements included. We have included a screenshot example of the end statements for reference. If you feel that a given

heading is not relevant to your paper, please nevertheless include the heading and explicitly state that it is not relevant to your work.

Because the schedule for publication is very tight, it is a condition of publication that you submit the revised version of your manuscript before 06-Feb-2020. Please note that the revision deadline will expire at 00.00am on this date. If you do not think you will be able to meet this date please let me know immediately.

on behalf of Dr Narayanan Srinivasan (Associate Editor) and Essi Viding (Subject Editor)
openscience@royalsociety.org

Associate Editor Comments to Author (Dr Narayanan Srinivasan):

The reviewers are reasonably satisfied with the revised manuscript. One reviewers still has some comments. Please address those comments and submit the final version.

Reviewer comments to Author:

Reviewer: 1

Comments to the Author(s)

I thank the authors for responding to all of my comments; I think that this is really interesting work. Keep it up!

Reviewer: 2

Comments to the Author(s)

Overall, I think the authors have dealt with my comments reasonably well. There are just a few outstanding issues that need to be addressed before this work can be published.

Kind regards.

Comments

For the habituation effects, please indicate the size of this effect, so that it can be compared to the other effects you report (e.g. for study 1, there is no data available to make this comparison). Also, I am not sure the randomised order of treatments ensures that habituation does not affect your findings, seeing that one random ordering is used in each study (e.g. in study 2 a new random order could have been created for each participant). A generalised linear model with predictors map type, trial number and possibly reversed crowd direction (study 3) and interaction terms would allow you to test this.

Line 32-33 on page 7: "...but this tendency is probably attributable to familiarity with the control interface in VR." Do you have any evidence to back this speculative statement up?

Comparing the results in tables 1 and 2, it seems that when using the ART ANOVA, you do not find a significant effect of map type on the number of errors.

Study 3: is it possible that the difference in the number of errors you find for the crowd movement treatment is entirely due to the outlier of trial number 9?

Page 11, lines 57-58: "Consistent with O'Neill [27], we found that simple maps helped people make faster [...] spatial decisions" given the small effect sizes and the non-significant differences in decision times, I don't think you can make this claim.

Author's Response to Decision Letter for (RSOS-191523.R1)

See Appendix B.

Decision letter (RSOS-191523.R2)

18-Feb-2020

Dear Mr Zhao,

It is a pleasure to accept your manuscript entitled "The interaction between map complexity and crowd movement on navigation decisions in virtual reality" in its current form for publication in Royal Society Open Science.

Kind regards,
Lianne Parkhouse
Royal Society Open Science
openscience@royalsociety.org

on behalf of Dr Narayanan Srinivasan (Associate Editor) and Essi Viding (Subject Editor)
openscience@royalsociety.org

Appendix A

Lianne Parkhouse
Editorial Coordinator
Royal Society Open Science

Zurich, 12 December 2019

Dear Editor Parkhouse,

We thank the editors and two reviewers for their detailed comments on our paper "The interaction between map complexity and crowd movement on navigation decisions in virtual reality." We have edited the manuscript accordingly and listed our responses in this Response Letter. We think that these changes have improved the quality of the manuscript substantially. We have also indicated where these changes were implemented in the manuscript.

Reviewer 1

Overall I am not terribly surprised by the results (in particularly study 1 and 2); given the very low error rate in both scenarios, I am wondering if the task itself was too easy. The simplified map essentially reduced the noise for the specific task at hand (e.g., when I need to find my gate, I do not need to know where the restrooms are). However, I wonder if the effects found here would hold up in a more difficult task (i.e., one that produces more errors at baseline). The authors could, for instance, task participants with finding an arbitrary item on the map, add time pressure (e.g., by using an evacuation scenario) or add visual noise to the virtual environment.

The reviewer raises an interesting point regarding how additional experimental conditions could have affected our results. We believe that three manipulations could be implemented in future work.

First, we could introduce time pressure in order to more realistically simulate the stress associated with finding a terminal gate. In the present study, we purposefully did not use an evacuation in order to keep the design simple, but we would expect this type of time pressure to increase the size of the effect of map complexity. This idea is consistent with previous research from our group (Moussaid et al., 2016), which has found that people are much more likely to follow others in an evacuation scenario with additional time pressure.

Second, adding visual noise (e.g., smoke from a simulated fire) would probably reduce the size of the effect of map complexity and increase following behaviour. Following vision-based models of collective motion (Moussaid, Helbing, & Theraulaz, 2011; Warren, 2018; Wirth & Warren, 2016), we would expect, especially with low visibility, for objects closer to the navigator to affect the navigator's behaviour more than objects further from the navigator. This principle should apply whether distances are considered metric or topological and whether these objects are signs or other people. We could also add visual noise by making the signs more complex (e.g., more icons or gates). By adding these details, the effect size would probably decrease. However, if the complexity of the two maps would be noticeably different, we could find an even larger effect of map type.

Third, task difficulty could be increased by having participants find the actual gate instead of turning left or right. Looking at the interaction between task difficulty and map complexity would be interesting because we could expect different types of interaction. On the one hand, the effect of map complexity could decrease with task difficulty because the task would be longer and involve more decisions along the route to the destination. On the other hand, the effect of map complexity could increase with task difficulty because the time required to find the destination is amplified by each decision along the route. We believe that addressing these issues would be beyond the scope of the present paper, but we have added a paragraph in the Discussion section that addresses these points (see page 12).

I am a little worried about the ecological validity of the virtual environment. I appreciate that the authors created a very reduced and controlled virtual environment; however, adding a little more complexity to the virtual environment so that it is closer to an actual terminal would probably be helpful. For instance, signage (which could be a confounding factor) is completely missing in the virtual environments. The authors could consider, recreating experiment 1 in a photomontage of O'Hare.

As noted above, we think that adding noise to the virtual environment, such as with additional signs on the ceiling and walls, could decrease the effect of map complexity that we found, especially if these signs are irrelevant to the task. However, as the reviewer mentions, if we added signs that were relevant to the task, this could have confounded our manipulation of map complexity. In general, we think that our environment was appropriate because it allowed us to isolate these effects in a reproducible way. Whether or not these effects would occur in a more realistic environment is an interesting topic for future study. We discuss this issue in the last paragraph on page 12 in the Discussion section.

Even the complex map that the authors presented appears simpler than the actual 3D terminal map shown Fig 1a. Did the authors consider comparing 3D maps to 2D maps?

Our complex maps are slightly simpler than the original map from O'Hare, including its dimensionality. We think that a comparison between 3D and 2D maps for navigation decisions would be interesting. Indeed, Hegarty and colleagues (2009) have found that people tend to prefer 3D maps but that 3D maps may hinder performance on spatial tasks. In addition, the original maps from O'Hare represent multiple floors that we did not have in the virtual environment. Without these additional floors, the maps look 3D but do not provide any additional information. We now address this concern in the Materials subsection of Study 1 on page 3.

Would it be realistic to expect generic airport maps to be simplified? The reason for their complexity, I assume, is to provide a flexible tool so that users can find a number of different locations (e.g., restaurants, terminals and restrooms). What would be the real life implications of the findings? Would the authors suggest developing task specific maps (e.g., in a smartphone application)?

We agree with the reviewer that the real-world application of this study is not to simplify all map designs. When designers implement a new map for pedestrians, they should always consider other aspects such as the amount and variety of the information presented on the map. With the convenience and ubiquity of smartphones, maps in airports can be displayed with an interactive digital device on which information regarding other types of locations can be presented for different tasks (e.g., searching for the nearest bathroom, finding a restaurant). The findings from our study can inspire such map designs to be as simplified as possible when first presented to the users. We now briefly discuss this point on page 12 of the Discussion section.

For methodological reasons, the authors placed the participants behind the crowd in experiment 3. However, when navigating airports, we are rarely following a crowd; that is, the signal from the social influence is probably noisier than in the studies presented here. I believe that there is also some research that suggest that crowd density (i.e., the number and distance to neighbors in a crowd) influences the how strongly people are influenced by a crowd. A quick google scholar search only revealed the reference below, but there might be more out there that the authors want to discuss: T. D. Wirth and W. H. Warren, "The visual neighborhood in human crowds: Metric vs. Topological Hypotheses," *Journal of Vision*, vol. 16, pp. 982–982, 2016.

The reviewer raises an interesting point. Because Study 3 depended on the traces from Study 2, it would be difficult to manipulate the trajectories of the agents without diminishing the social

signal provided by the agents. It was necessary to create a social signal (albeit somewhat artificially) in order to test whether participants could pick up on this social signal. Although this social signal may be somewhat diluted in most real-world scenarios, we think that this was an important first step. Also, we have now added Wirth & Warren (2016) and Warren (2019) to the Introduction on page 2.

I really liked the normalized hesitation heat maps. Did the authors consider plotting difference maps? This might make a comparison between the conditions more intuitive. I am not sure about this though.

We like the idea of the difference map and plotted it in Figure R1 below. The subcaptions indicate the way the difference maps were generated. For example, in Figure R1a “Complex/Original - Simple/Original”, we subtracted the normalized hesitation points of the simple map group with original crowd movement from complex map group with the original crowd movement. As expected, the difference maps are consistent with the normalized density maps from the original manuscript because the locations of differences are similar to the most common hesitation points across all four maps. However, we think that the difference maps are somewhat less informative than the normalized density maps because dark blue always represents the absence of a hesitation in the normalized density maps but could be interpreted one of two ways for the difference maps (either no hesitation points or a difference of zero). For this reason, we believe that we should keep the original normalized density maps in the manuscript, but we are open to including the difference maps as an additional figure if requested.

Figure R1. Difference map of the normalised density maps of hesitation points.

The authors probably played around with the cell sizes of the density maps. These should be reported in the figure caption; and also a rationale for the specific cell size should be given.

First, it is worth noting that the scale of the density maps was unrelated to the calculation of the statistics based on Kernel Density Estimates (KDE) and described on page 8 in the Design subsection of the manuscript. However, we appreciate the reviewer's concern and have replotted these density maps with a scale of 0.66m x 0.66m (see Figure R2), which corresponds to the average young adult's step length (Noboru, and Nagasaki, 1998). Compared to the original density maps, we do not see any substantial differences in the locations of normalized hesitation maps. Because of the similarity between these two sets of density maps, we will leave the choice of density maps to the editor's and reviewers' discretion.

Figure R2. Comparisons of different cell sizes of density maps

I was wondering about the consequences of identifying hesitation points. These could create unexpected obstacles and impede crowd flow in general.

We thank the reviewer for pointing this out, and we have added these points to the Discussion section on page 12. Specifically, we discuss our results relative to (Zou et al., 2019; Kirchner et al., 2003; Khan et al., 2017) to highlight that hesitations are related to collisions, decreases in travel speed and congestion. Previous research has already shown that hesitations by pedestrians can increase the probability of collisions and decrease the moving speed of a crowd (Zou et al., 2019; Kirchner et al., 2003). Such involuntary slowing down of crowds can create unexpected obstacles and lead to congestion (Khan et al., 2017). This finding further confirms the necessity of the simpler map design.

Did the authors consider analyzing the heading data that could be derived from the mouse movement?

We agree with the reviewer that heading data is an interesting factor to reveal how often people change their decision during navigation. However, we believe that the measure of hesitation points is more likely to have direct consequences for crowd flow in public spaces. While heading data might capture hesitation indirectly, it may also indicate when participants are swerving to avoid collisions. In contrast, hesitation points represent the participants' actions of either slowing down or stopping between two recorded data points. Within the visual catchment area of the sign, hesitation most likely represents confusion caused by the sign. We believe that hesitation points are a better representation of how people changed their mind.

Reviewer: 2

Just a note on page 2: “...likely to herd as a crowd...”: it has recently been suggested that several terms, including “herding” are not sufficiently precise. Seeing that you refer to “following behaviour” in the same sentence, why not just use the latter, more precise, phrase. See: Haghani, M., Cristiani, E., Bode, N. W., Boltes, M., & Corbetta, A. (2019). Panic, irrationality, and herding: three ambiguous terms in crowd dynamics research. *Journal of advanced transportation*, 2019.

We agree with the reviewer and have changed the word from “herding” to “following behaviour” throughout the manuscript. In addition, we define the term “following behaviour” as conforming to the actions of the crowd (Haghani et al., 2019) in the Introduction section of the manuscript on page 2.

Page 2: “For example, Moussaïd and colleagues [7] validated a multi-user VR framework using a study of collective navigation behaviour and herding during a stressful evacuation.” The comparisons of the VR framework in [7] focussed on operational level behaviour (passing another person in a corridor and flow depending on bottleneck width), aspects that are dependent on simple physics in the VR (walking speed, size of avatars, etc). The match between VR and experiments was not exactly perfect. So what do you mean by “validated” here?

Originally, we used the word “validate” to suggest that this same VR setup has been successfully used in similar studies. However, we agree with the reviewer that the word “validation” is imprecise in this context. We have now changed this word to “implemented”.

Study 1: each participant performed 24 trials. You should test for habituation effects in your statistical analysis.

We tested for habituation effects using the linear contrast from a repeated measures ANOVA. This contrast revealed a significant linear effect of trial on time required to final decision, $F(1,144) = 35.966$, $MSE = 115.362$, $p < .001$. We now report this effect in the Results section of Study 1 on page 5. It is worth noting that this should not affect our interpretation of the difference between simple and complex signs because the order of trials was randomized for each participant.

In the caption on figure 3, it would be useful to indicate the number of data points making up the bar charts displayed (145x12?).

We have also added the number of data points to the caption of Figure 3. The text now reads “There are 145 data points in total represented by each bar.”

Did you check the t test assumptions hold? The size of the difference in the number of clicks in fig. 3 is very small. It would be good to mention this.

We have tested the assumptions of a t-test for the equality of variances using Levene's test. These tests did not reveal any significant differences between the simple and complex groups' variances for time to final click ($p = 0.611$), for percentage of error ($p = 0.071$), or for number of clicks ($p = 0.959$). We have also calculated Cohen's d as a measure of effect size for Study 1. We found a small to medium effect size for accuracy $d = 0.205$ and a very small to small effect size for number of clicks $d = 0.113$. We have added the Levene's tests and effect sizes to the Results section of Study 1 on page 5.

Study 2: again, it would be good to explicitly mention the small effect size in the difference in time taken to complete trials.

We have calculated the effect size for Study 2 and found a very small to small effect for the time required to complete each trial ($d = 0.131$). We have added the effect sizes to the Results section of Study 2 on page 6.

Also, please discuss the habituation effect (reduction in time taken with trial number) which seems to be visible in fig. 4.

We have tested habituation for Study 2 (using the linear contrast from a repeated measures ANOVA) and found a significant linear effect of trial on time required to complete each trial, $F(1,28) = 16.305$, $MSE = 36.37$, $p \leq .001$. We have reported this effect in the Results section of Study 2 on page 7. Notably, this habituation does not affect our interpretation of the difference between simple and complex signs because the order was randomly predetermined and evenly distributed over time for all participants.

Finally, it wasn't clear to me if participants all had the same goal?

We have added one sentence in the Procedure section of Study 2 on page 6 to clarify that all participants were given the same gate for each trial. The part now reads "... participants were not instructed as to whether the other participants were given the same gate as a goal. All participants were given the same gate for each trial."

Study 2: you only use one group of participants. This is a major limitation of your study as it is not clear how your results would vary if another group of participants performed the experiment. You must state this limitation explicitly somewhere in the text.

We agree with the reviewer and have added this limitation in the Discussion section on page 11. "Although we only tested for following behaviour using one group of participants, this finding is consistent with previous research that found that participants tended to hesitate before responding to a similar conflict between the direction indicated by an computer-controlled avatar and the actual direction of an emergency exit (Kinatered et al., 2014)."

Captions of table 1 and 2: please indicate which study the results presented in the table are from.

We have now indicated the number of the study in the Table captions.

Correction

When calculating the effect size for the Study 2, we noticed an error in the calculation of mean time taken for the Complex and Simple map groups. The new calculation revealed that their difference was not significant. However, because Study 2 was designed to collect crowd trajectories for Study 3, this corrected mean time from Study 2 does not influence the findings or interpretation of Study 3. We have corrected this error in the revised manuscript, and we apologize for this mistake.

Yours sincerely, on behalf of all authors,

Hantao Zhao

References

Kirchner, A., Nishinari, K., & Schadschneider, A. (2003). Friction effects and clogging in a cellular automaton model for pedestrian dynamics. *Physical review E*, 67(5), 056122.

Zou, B., Lu, C., Mao, S., & Li, Y. (2019). Effect of pedestrian judgement on evacuation efficiency considering hesitation. *Physica A: Statistical Mechanics and its Applications*, 122943.

Haghani, M., Cristiani, E., Bode, N. W., Boltes, M., & Corbetta, A. (2019). Panic, irrationality, and herding: three ambiguous terms in crowd dynamics research. *Journal of advanced transportation*, 2019.

Hegarty, M., Smallman, H. S., Stull, A. T., & Canham, M. S. (2009). Naïve cartography: How intuitions about display configuration can hurt performance. *Cartographica: The International Journal for Geographic Information and Geovisualization*, 44(3), 171-186.

Kinateder, M., Müller, M., Jost, M., Mühlberger, A., & Pauli, P. (2014). Social influence in a virtual tunnel fire— influence of conflicting information on evacuation behavior. *Applied ergonomics*, 45(6), 1649-1659.

Moussaïd, M., Helbing, D., & Theraulaz, G. (2011). How simple rules determine pedestrian behavior and crowd disasters. *Proceedings of the National Academy of Sciences*, 108(17), 6884-6888.

Sekiya, N., & Nagasaki, H. (1998). Reproducibility of the walking patterns of normal young adults: test-retest reliability of the walk ratio (step-length/step-rate). *Gait & posture*, 7(3), 225-227.

Khan, S. D., Tayyab, M., Amin, M. K., Nour, A., Basalamah, A., Basalamah, S., & Khan, S. A. (2017). Towards a Crowd Analytic Framework For Crowd Management in Majid-al-Haram. *arXiv preprint arXiv:1709.05952*.

Wirth, T., & Warren, W. (2016). The visual neighborhood in human crowds: Metric vs. Topological Hypotheses. *Journal of Vision*, 16(12), 982-982.

Warren, W. H. (2018). Collective motion in human crowds. *Current directions in psychological science*, 27(4), 232-240.

Appendix B

Andrew Dunn
Editorial Coordinator
Royal Society Open Science

Zurich, 14 February 2020

Dear Editor Dunn,

We thank the editors and two reviewers for their detailed comments on our paper "The interaction between map complexity and crowd movement on navigation decisions in virtual reality." We have edited the manuscript according to the comments of the second reviewer and listed our responses in this Response Letter. We think that these changes have improved the quality of the manuscript. We have also indicated where these changes were implemented in the manuscript.

Reviewer: 2

1- For the habituation effects, please indicate the size of this effect, so that it can be compared to the other effects you report (e.g., for study 1, there is no data available to make this comparison).

We have now calculated the effect sizes for the linear contrast from Studies 1 and 2. Our estimate of Cohen's d was converted from η^2 using the following equation, assuming intermediate dispersion of the trial means (Cohen, 2013, pg. 279.):

$$d = 2 * f * \sqrt{(3 * (k - 1) / (k + 1))}$$

For Study 1, we found a large-sized habituation effect ($d=0.830$), and for Study 2, we found a huge-sized habituation effect ($d=1.969$). These details are now added to the Results of Studies 1 (page 6) and 2 (page 7).

2- Also, I am not sure the randomised order of treatments ensures that habituation does not affect your findings, seeing that one random ordering is used in each study (e.g. in study 2 a new random order could have been created for each participant). A generalised linear model with predictors map type, trial number and possibly reversed crowd direction (study 3) and interaction terms would allow you to test this.

Because participants were simultaneously immersed in the same environment for each trial of Study 2, the order of the experimental conditions could not have been randomised differently for each participant. Nonetheless, we now account for the possible effect of habituation over trials by including trial number as a fixed factor in a linear mixed effects model. For Study 2, we found that this approach actually produces a significant effect of map type on the time required to complete each trial in the predicted direction, $F(1,405) = 49.149$, $MSE=8.613$, $p < .001$.

For Study 3, we conducted linear mixed effects models with crowd movement, map type, and trial number as fixed factors separately for the four dependent variables (i.e., number of errors, time, number of hesitation points, and number of hesitation points within the VCA). We repeated these analyses for all of the 32 trials together and again for the 31 trials excluding the trial (#9). Because trial number could not have been manipulated independently from map type and crowd movement, we could not include any interaction terms in these models. The results of these models are summarized in Table 1 below. Consistent with our previous analyses of habituation, we found effects of trial number on all four dependent measures when trial #9 is included. When trial #9 is excluded, the habituation effect remains only for time. In addition, we found effects of crowd direction on time with or without trial #9, and an effect of map type on time when trial #9 is excluded.

Overall, these results suggest that habituation over trials may help explain some of the effects we originally obtained, but it is difficult to directly compare linear mixed effects models with three fixed factors that were not manipulated factorially to 2x2 ANOVAs that include interactions. Given that our primary hypothesis was the interaction between crowd movement and map type, we think that the original analyses are the most appropriate. However, we agree with the reviewer that additional studies could be designed to randomise the order of trials after the probe trial #9. We now note this possible limitation in the Discussion (page 12) sections.

Table 1. Comparison of P value from generalized linear model from Study 3, with and without Trial 9

		P = (With trial 9)	P = (Without trial 9)
Number of Errors	Crowd	0.498	0.454
	Map	1.000	1.000
	Trial	< 0.001	0.076
Time	Crowd	0.002	< 0.001
	Map	0.057	0.040
	Trial	< 0.001	< 0.001
Number of Hesitation Point	Crowd	0.070	0.059
	Map	0.103	0.089
	Trial	< 0.001	0.053
Number of Hesitation Point inside VCA	Crowd	0.144	0.132
	Map	0.122	0.111
	Trial	< 0.001	0.052

3- Line 32-33 on page 7: “...but this tendency is probably attributable to familiarity with the control interface in VR.” Do you have any evidence to back this speculative statement up?

We have edited this sentence on page 7 to clearly state that our interpretation is somewhat speculative and based on previous literature (Rozado, 2013; Santos et al., 2009). We agree with the reviewer that we could specifically test for this possibility in future studies by directly manipulating training time with the control interface.

4- Comparing the results in tables 1 and 2, it seems that when using the ART ANOVA, you do not find a significant effect of map type on the number of errors.

We agree the reviewer and have revised the sentence on page 8 from “Both parametric and non-parametric analysis revealed a main effect of crowd movement and map complexity on number of errors” to “Both parametric and non-parametric analysis revealed a main effect of crowd movement on number of errors. The effect of map types on the number of errors was only significant in the parametric analysis.”

5- Study 3: is it possible that the difference in the number of errors you find for the crowd movement treatment is entirely due to the outlier of trial number 9?

We conducted the 2x2 ANOVAs with and without trial #9 and still found a significant interaction between the effects of map type and crowd movement on time, $F(1,1)=5.156$, $MSE=3.081$,

p=0.031. However, we noticed that eliminating trial #9 removes all of the other significant effects (see Table 2). Ideally, we would be able to exclude trials 1 through 9 in order to focus on trials in which participants were aware of the possible conflict between the direction indicated by the map and the direction indicated by the crowd. This would require additional trials with the original crowd in order to balance the various experimental conditions. We now note this possible limitation in the Discussion (page 12) sections.

Table 2. Comparison of P value of Two-way ANOVA results for all four dependent variables from Study 3, with and without Trial 9

		P = (With Trial 9)	P = (Without trial 9)
Number of Errors	2X2 interaction	0.003	0.813
	Crowd	0.038	0.074
	Map	0.005	0.787
Time	2X2 interaction	0.007	0.031
	Crowd	0.958	0.379
	Map	0.150	0.633
Number of Hesitation Point	2X2 interaction	0.176	0.821
	Crowd	0.207	0.812
	Map	0.321	0.847
Number of Hesitation Point inside VCA	2X2 interaction	0.177	0.583
	Crowd	0.189	0.900
	Map	0.117	0.400

6- Page 11, lines 57-58: “Consistent with O’Neill [27], we found that simple maps helped people make faster [...] spatial decisions” given the small effect sizes and the non-significant differences in decision times, I don’t think you can make this claim.

We agree with the reviewer and have edited this sentence accordingly. It now reads, “Consistent with O’Neill [27], we found that simple maps may help people make more accurate spatial decisions.”

Correction

While addressing Reviewer #2's comments on the analyses of hesitation points within the VCA, we found a mistake in the calculation of the VCA. We have now corrected this mistake and updated the ANOVAs reported in Table 2, Figure 6 and the analyses of the density maps using KDE in the manuscript. Unfortunately, the interaction from the 2x2 ANOVA for hesitation points within the VCA is no longer significant, but the pattern of results from the KDE analyses remains the same. We have incorporated these changes in the results and the discussion section of the manuscript.

Yours sincerely, on behalf of all authors,

Hantao Zhao

References

Cohen, J. (2013). *Statistical power analysis for the behavioral sciences*. Routledge.

Rozado, D. (2013). Mouse and keyboard cursor warping to accelerate and reduce the effort of routine HCI input tasks. *IEEE Transactions on Human-Machine Systems*, 43(5), 487-493.

Santos, B. S., Dias, P., Pimentel, A., Baggerman, J. W., Ferreira, C., Silva, S., & Madeira, J. (2009). Head-mounted display versus desktop for 3D navigation in virtual reality: a user study. *Multimedia tools and applications*, 41(1), 161.